# Impact of Repetitive Transcranial Magnetic Stimulation on Cognitive and Psychiatric Dysfunction in Patients with Fibromyalgia: A Double-Blinded, Randomized Clinical Trial

**DOI:** 10.3390/brainsci14050416

**Published:** 2024-04-24

**Authors:** Marwa Y. Badr, Gellan K. Ahmed, Reham A. Amer, Hend M. Aref, Rehab M. Salem, Heba A. Elmokadem, Eman M. Khedr

**Affiliations:** 1Department of Neuropsychiatry, Faculty of Medicine, Tanta University, Tanta 31511, Egypt; drmoroneuro@yahoo.com (M.Y.B.); rehamamer79@yahoo.com (R.A.A.); psydrhendaref@gmail.com (H.M.A.); 2Department of Neurology and Psychiatry, Faculty of Medicine, Assiut University, Assiut 71526, Egypt; gillankaram@aun.edu.eg; 3Department of Physical Medicine, Rheumatology and Rehabilitation, Faculty of Medicine, Tanta University, Tanta 31511, Egypt; rehab.salem@med.tanta.edu.eg (R.M.S.); heba.elmokadem@med.tanta.edu.eg (H.A.E.)

**Keywords:** fibromyalgia, cognitive impairment, psychiatric disorders, neuropsychological tests, rTMS

## Abstract

Few randomized controlled trials have reported that repetitive transcranial magnetic stimulation (rTMS) has controversial results for managing multiple domains of fibromyalgia-related symptoms. This work aimed to evaluate the effect of low-frequency rTMS over the right dorsolateral prefrontal area (DLPFC) on the Fibromyalgia Impact Questionnaire (FIQ) concerning psychiatric and cognitive disorders. Forty-two eligible patients with fibromyalgia (FM) were randomized to have 20 sessions of active or sham rTMS (1 Hz, 120% of resting motor threshold with a total of 1200 pules/session) over the right DLPFC. All participants were evaluated at baseline, post sessions, and 3 months after sessions with the FIQ, Hamilton depression, and anxiety rating scales (HDRS and HARS), Montreal Cognitive Assessment (MoCA), Rey Auditory Verbal Learning Test (RAVLT), Tower of London test (TOL), the Trail Making, and Digit Span Tests. Both groups showed improvement in most rating scales at 1 and 3 months follow-up, with greater improvement in the active group, with significant correlation between FIQ cognitive rating scales, including RAVLT and TOL. Twenty sessions of low-frequency rTMS over the right DLPFC can improve FIQ scores regarding the psychiatric and cognitive symptoms of medicated patients with FM to a greater extent than sham. Changes in RAVLT and TOL correlated with changes in FIQ results.

## 1. Introduction

Fibromyalgia syndrome (FMS) is a systemic and multifactorial disease of unknown etiology affecting approximately 0.5–6.6% of the world population, with symptoms not only somatic (muscular and joint) but also within the cognitive and psychiatric spheres (anxiety and depression) [1]. Fibromyalgia is a chronic pain syndrome and frequently affects women. In FM, widespread pain (≥3 months) is the dominant feature and incorporates a wide variety of symptoms, including fatigue, sleep disorders, and cognitive and mood disorders. Although the exact cause of FM is still unknown, aberrant pain perception is usually explained by central sensitization [2] with the consequence that individuals with FM experience nociceptive hypersensitivity to non-noxious stimuli [3,4].

Recently published diagnostic criteria for FMS indicate that chronic pain is associated with cognitive impairment, aberrant neuroplasticity, and neurochemical changes [5,6]. Additional psychological manifestations such as depression, anxiety, and personality disorders are also prevalent in patients with FM, all of which could influence pain severity and contribute to poor overall health [7].

A noninvasive brain stimulation (NIBS) technology called repetitive transcranial magnetic stimulation (rTMS) uses electromagnetic fields to modify activity in specific areas of the cerebral cortex. In the last 10 years, it has been widely used to carry out brain connectivity and neuromodulation studies. It has been shown to be beneficial for treating neuropathic pain [8,9], especially in cases where pharmacological treatment has failed, as well as for cognitive impairment in neurodegenerative diseases such as Alzheimer’s, Parkinson’s, and vascular dementia [10,11]. rTMS has been used to treat psychiatric disorders such as OCD and treatment-resistant depression (TRD) [12,13]. It may also be viewed as an adjunctive therapy to the standard course of care in a number of conditions [14,15]. Dorsal prefrontal cortex (DLPFC) stimulation alone appears to have antidepressant effects on psychosocial variables and cognition, according to evidence-based guidelines for rTMS [16].

Additionally, the rTMS of DLPFC plays a special executive attention role in actively preserving access to stimulus representations and objectives in environments with plenty of distraction such as those of team sports. Volleyball is a team sport in which the attention and coordination components are essential for achieving performance [17,18]. They concluded that the rTMS of DLPFC improved reaction time (RT) and the number of correct answers, and cortical excitability in volleyball players after active rTMS can increase coordination performance when the velocity of the execution is high.

Many studies have evaluated the efficacy of rTMS for relieving pain in FM; however, few have evaluated the efficacy of rTMS in modulating the associated symptoms of FM, such as fatigue and disorders of sleep, cognition, and mood, nor have they examined the relationship between changes in pain and cognitive symptoms.

The European expert group recommended rTMS as a treatment level B for fibromyalgia in 2020 [19]. The main approach is to reduce the activity of pain-related areas by modifying cortical excitability [20]. The mechanisms of rTMS include enhancing endogenous opioid release and producing neuroplastic changes in pain circuits [20,21,22]. Depending on the cortical area stimulated and the frequency of rTMS, the therapeutic effects can vary [23]. The primary motor cortex (M1) and dorsolateral prefrontal cortex are the main brain areas that have been targeted by rTMS to treat pain [24].

A meta-analysis that included ten randomized trials concluded that HF rTMS over M1 reduced visual analog pain ratings in individuals with FM [25] for one to four weeks after treatment. In contrast, another study found that LF rTMS over the DLPFC provided effective pain relief [26]. Since there is no consensus on the optimal location and stimulation frequency of rTMS to treat pain [27], we chose in the present study to use low-frequency rTMS over the left DLPFC and evaluated its effect on FIQ regarding cognitive and psychiatric abnormalities in cases with FM.

## 2. Materials and Methods

This was a prospective double-randomized controlled trial that was conducted in Tanta and Assiut University hospitals during the period from 1 October 2022 to the end of March 2023. Sixty-four patients with fibromyalgia were recruited from the outpatient clinics of Physical Medicine, Rheumatology and Rehabilitation and Neuropsychiatry Departments, Tanta, and Assiut University Hospitals. The diagnosis of fibromyalgia was made according to the American College of Rheumatology criteria [28,29]. Inclusion criteria: Participants aged ≥ 18 years who have been on their medication for the last 3 months without any improvement and express dissatisfaction. These medical treatments continued without changes throughout the time of the study. Exclusion criteria: (1) Inflammatory rheumatic disease, autoimmune disease, or other painful disorders. (2) Any uncontrolled clinical disease (such as thyroid, cardiovascular, pulmonary, hematological, or renal disease) that affects cognition. (3) Pregnancy, lactation. (4) Contraindications for rTMS (a history of seizures, brain trauma, brain surgery or intracranial hypertension, a pacemaker, or other metallic implants). (5) Past history of other neurological disorders (neurodegenerative diseases, dementia), primary psychiatric disorders (psychosis or major depression (17-item Hamilton Depression Rating Scale > 23), or drug treatments that affect cognition. (6) Inability to cooperate with the questionnaire survey or the patient refuses to participate in the study. 

Sixty-four patients with fibromyalgia were recruited. Twenty-two cases were eliminated because they did not fit the inclusion and exclusion criteria: eight of them suffered from systemic illnesses, five patients refused to participate, four were not candidates for magnetic stimulation, and five cases involved primary psychiatric conditions. Forty-two cases met the eligibility criteria and were divided into the active rTMS group and the sham rTMS group in a 1:1 ratio. Due to a headache, one patient from each group was unable to finish the follow-up study; the other twenty cases in each group finished the investigation. (see Figure 1 for the flowchart).

Following a detailed clinical assessment, patients completed the following assessments: 

Fibromyalgia Impact Questionnaire (FIQ) [30]. The FIQ is a brief 10-item self-administered instrument that takes approximately 5 min to complete. It has been designed to measure physical functioning, work status, depression, anxiety, sleep, pain, stiffness, fatigue, and wellbeing. The average FM patient scores about 50, while severely afflicted patients usually score 70 plus. 

The MoCA [31]: We applied the validated Arabic version. It takes 10 to 12 min to complete and can detect mild cognitive impairment. It does so based on 11 questions that evaluate seven domains of cognitive function (executive and visuospatial function, naming, attention, language, abstraction, delayed recall, and orientation). The MoCA has a maximum score of 30, and anything below 24 is a sign of cognitive impairment [32,33]. 

Executive function domains were measured using the Rey Auditory Verbal Learning Test [RAVLT]) [34]. The RAVLT is a neuropsychological assessment designed to evaluate verbal memory. Five presentations of a 15-word list are given, each followed by attempted recall. This is followed by a second 15-word interference list (list B), followed by a recall of list A. Delayed recall and recognition are also tested. The number of correct words was summed for the Total Recall score (range = 0–75). The Delayed Recall score is the number of correct words recalled after a 30 min delay (range = 0–15). 

The Tower of London test (TOL) for executive planning proficiency [35]: The TOL is considered a general measure of visuospatial problem-solving, and more specifically of planning, related to the classic problem-solving puzzle known as the Tower of Hanoi. Scores are calculated as follows: (1) Total score equals the sum of scores on each trial. (2) Solution time (seconds) equals the sum of the time spent on each item between the instruction and the first movement. (3) Execution time (seconds) is the difference between the sum of the total time spent on each item and the planning time.

The Trail Making Test (TMT) for visual attention and task switching [36]: It has two parts, in which the subject is instructed to connect a set of 25 dots as quickly as possible while maintaining accuracy. The test can provide information about visual search speed, scanning, speed of processing, mental flexibility, and executive functioning [37]. 

The Digit Span Test (DS) for verbal short-term and working memory: DS is a measure of verbal short-term and working memory that can be used in two formats, Forward Digit Span and Reverse Digit Span [38]. 

Patients were also assessed psychiatrically using Arabic versions of the Hamilton Depression Rating Scale [39] (HDRS). The original version contains 17 items (HDRS 17) pertaining to symptoms of depression experienced over the past week. The Hamilton Anxiety Rating Scale (HARS) [40] consists of 14 items, each defined by a series of symptoms, and measures both psychic anxiety (mental agitation and psychological distress) and somatic anxiety (physical complaints related to anxiety). We used the Arabic version [41].

All psychometric tests were carried out by 2 qualified neuropsychiatric consultants, and the meaning of each score was taken.

Randomization:

Participants were randomized into active and sham treatment groups using sealed, serially numbered envelopes. Prior to enrollment, participation-related counseling was given. Forty-two fibromyalgia patients who had been treated with appropriate medical treatment were included.

Procedure of repetitive transcranial magnetic stimulation (rTMS):

Resting motor threshold (rMT) was determined for each patient by moving the coil over the right motor hand cortex in order to detect the smallest possible intensity needed to produce a visible movement of the left first dorsal interosseous muscle (FDI) of the hand [42]. For determination of the stimulation site of the right DLPFC, it was specifically identified as the region located 5 cm in front of the ideal position for motor threshold production in the FDI [43] (see Figure 2 for the location and position of the coil). rTMS was delivered by a qualified expert. 

Experimental design: Each patient received 20 trains of low-frequency (1 Hz) rTMS at 120% of rMT of the left FDI; each train consists of 60 s with a 45 s intertrain interval, giving a total of 1200 pulses for each session (5 sessions/week for 4 consecutive weeks) (see Figure 3). The rTMS sessions were carried out using MAGSTIM equipment (Company Limited, Whitland, Wales, UK) connected to a 7 cm diameter figure of eight coil [44]. 

The sham rTMS group underwent the same protocol as the active group, except that stimulation of the DLPFC was delivered with the coil held perpendicular to the scalp. The researcher administering rTMS did not interact with the patients, maintaining a double-blind protocol.

Follow-up of the patients:

Each participant was reassessed post session and 3 months after the end of rTMS treatment. They were asked to report side effects and any inconvenience during or after the procedure. The investigator responsible for the assessment was unaware of the type of stimulation (active or sham) that the patient had received.

Outcomes:

Primary outcomes were the changes from pre–post-treatment sessions and pre–post 3 months after treatment sessions in FIQ. 

Secondary outcomes were the changes from pre–post-treatment sessions and pre–post 3 months after sessions in Hamilton scales and cognitive scales.

Statistical Analysis:

Data were analyzed utilizing the SPSS V17. Mann–Whitney nonparametric tests were applied to compare demographic and clinical data between groups. For qualitative information, a comparison between the two groups was performed utilizing the Chi-square test. Score differences from baseline were measured at one and three months following treatment for each rating scale. The difference between groups was assessed using Mann–Whitney nonparametric U tests. The difference between the mean changes (pre–post 1 and 3 months after the end of sessions) of the two groups is expressed in standard deviation units. Correlation analysis was performed using Spearman’s correlation test to evaluate the correlation of the mean changes in different scales with each other. Multivariate linear regression was performed to assess possible risk factors of the mean changes in FIQ with different scales. *p*-values of <0.05 were considered significant.

## 3. Results

There were no significant differences between groups at baseline in age, sex, education years, duration of illness, FIQ, HDRS and HARS, MoCA, or other rating scales (Table 1). 

Table 2 shows the different pharmacological medications among the studied groups. There were no significant differences between the groups.

Table 3 and Figure 4 show that the changes (pre–post 1 month and pre–post 3 months) in FIQ, HDRS, and HARS scores were significantly greater in the active group versus the sham group. The advantage of active treatment over sham at 1 and 3 months post treatment was 12.35 and 8.25 points for the FIQ; 3.75 and 4.25 points for HDRS; and 4.75 and 3.4 points for HARS at 1 month and 3 months, respectively.


**Neuropsychological Test Results:**


Table 4 shows the changes in different cognitive test scores at 1 and 3 months post treatment of the studied groups. At 1 month and 3 months, there were significantly greater improvements in all scores, apart from the TOL test at 1 and 3 months and MoCA at 1 month in the active group compared with the sham group.

Table 5 shows the results of the Spearman correlation analysis. There were significant correlations between changes in FIQ and Rey Auditory Verbal Learning Test (RAVLT) (sum of trials 1–5 immediate recall) and the Tower of London test (TOL) (total score), but only at 1 month post treatment.

## 4. Discussion

Most authors nowadays consider FM to have a central origin, caused by altered signal processing in brain circuits involved in the perception of pain [45]. However, although pain is the predominant symptom, other symptoms include fatigue, mood problems, and neurocognitive difficulties [46].

The main findings of this study are (1) a significantly greater improvement (pre–post change) in total FIQ, HDRS, and HARS in the active treatment group compared with the sham group for short- and long-term follow-up; (2) a greater improvement in different cognitive scales, particularly, MoCA, RAVLT, and TOL; and (3) a significant correlation between the changes in FIQ on one hand and RAVLT and TOL on the other hand.

The significantly greater improvement (pre–post change) in total FIQ score in the active treatment group compared with the sham group at both one and three months after the treatment sessions ended is consistent with the findings of Yağcı et al. [47], who reported a statistically significant improvement in FIQ scores on the 10th day and 1st and 3rd months after active rTMS treatment compared with sham. Furthermore, a review of 11 articles mentioning the FIQ score [48,49,50,51,52] indicates that FIQ scores after treatment were significantly lower after active rTMS (DLPFC) than sham. In a previous study by Tanwar et al. [53], the rTMS of the right DLPFC improved quality of life as evaluated by the FIQ, suggesting that stimulating the right DLPFC may positively impact the limbic system, particularly the right medial temporal cortex (involved in regulating the emotional aspects of pain) and the superior temporal sulcus (involved in social cognition and perception that underlies social functioning and quality of life), which also have connections with the limbic system.

Additionally, in the present study, active rTMS over DLPFC improved scores on the HDRS and HARS more greatly than that of the sham group. This is in agreement with Lee et al., who investigated the effect of 10 sessions of low-frequency right DLPFC rTMS (1 Hz, 110% motor threshold) on mood in a small, randomized controlled trial of 15 women with FM. They reported that depression and anxiety symptoms were significantly decreased from baseline after 1 month of treatment [48]. Our results also align with those of three other publications that found that applying rTMS over the DLPFC in FM patients led to a significant reduction in HDRS and HARS scores compared with patients receiving only medical treatment [54,55,56]. Previous studies have shown that the DLPFC is connected to the anterior insula and amygdala, regions linked to anxiety and depressive symptoms, which may explain our findings.

The second finding In the present study is the result of cognitive outcomes, as we selected a number of different cognitive scales to study the impact of rTMS on cognition at 1 and 3 months follow-up. The different cognitive scales include the MoCA, which is a concise cognitive screening test that evaluates eight cognitive domains (attention, executive function, calculation, language, working memory and recall, abstraction, orientation, and visuospatial processing) [57], and we also examined several specific domains of cognitive assessment: (1) executive function with the RAVLT and TOL; (2) working memory, including the Digit Span Test (forward and backward) for assessment of verbal short-term and working memory; and (3) visual attention with the TMT. In general, working memory, attention, conflict resolution, and verbal fluency have all been reported to be impaired in FM [58,59]. We targeted the DLPFC since it is a crucial brain region involved in regulating emotions like pain. But since it also plays a key role in various cognitive functions such as cognitive flexibility, working memory, and organization, we had expected that rTMS would also influence these symptoms. Indeed, the rTMS of DLPFC has been reported to address cognitive difficulties in healthy young individuals [60,61]. However, despite this, this present study on FM found that active stimulation had an effect on these measures of cognitive function at 1 month (RAVLT and TOL only) and had a minor effect at 3-month follow-up compared with the sham. Supporting our results, Baudic reported that the rTMS of DLFPC had only a mild effect on cognitive function in patients with FM [62]. 

Possible mechanisms for the effects of rTMS on depression and anxiety in FM patients may involve top-down modulation by rTMS targeting the right DLPFC. The DLPFC is linked to other brain regions such as the bilateral amygdala and contralateral anterior insula, where heightened neuronal activity is shown in relation to depression symptoms [4]. This connection allows the DLPFC to possibly influence this network.

The enhancement in cognitive and mood functions following rTMS may also be linked to the connectivity between the right DLPFC and limbic system, specifically the right medial temporal cortex responsible for emotional pain modulation, as well as the superior temporal sulcus involved in social cognition and perception affecting social quality of life. Previous studies have identified neural connections between these regions and the limbic system [63,64]. 

The significant correlation between the changes In FIQ and cognitive enhancement changes and the absent correlation between FIQ and psychiatric score changes may suggest that we have an effect on FIQ and some cognitive scores with a good relationship with each other, but not on psychiatric scores. Perhaps, it is not surprising that the mechanisms of rTMS are different for each domain.

Interestingly, in the present study, we applied more rTMS sessions than any other such study has to date, which enhanced the persistence of improvement at 3 months follow-up. However, the improvement observed in different rating scales among sham groups is not unusual in clinical trials, particularly those involving subjective measures or conditions with a strong psychological component. It may be related to the following: The test–retest effect, which refers to the phenomenon where participants’ scores on a measure may improve or change simply due to the repeated administration of the same test or measure over time, regardless of any intervention or treatment. This effect can occur due to several reasons: the practice effect, as participants may become more familiar with the test leading to improved performance or different responses on subsequent administrations; increased self-awareness, as the act of completing the same measure multiple times; and the familiarity with the testing process, as the participants may feel more comfortable and less anxious during subsequent testing sessions, which could impact their responses or symptom reporting [65,66,67].


**The Strengths of this Study:**


The number of treatment sessions (20 sessions) is greater than any previous study that was conducted on FM patients. The assessment scales of this study, including the impact of fibromyalgia on quality of life (FIQ), and both psychiatric and different cognitive assessments for a somewhat long time (3 months), were carried out by two qualified neuropsychiatric consultants blind to the type of treatment sessions and the meaning of each score taken.


**Limitations of the Study**


The small number of patients studied and the short duration of the follow-up were major limitations of this study. We recommend further studies with a larger sample size and long-term follow-up.

## 5. Conclusions

Patients diagnosed with FM had a broad spectrum of symptoms, including fatigue and psychiatric and cognitive abnormalities. The application of 20 sessions of 30 min of low-frequency rTMS over the right DLPFC proved to be effective and safe for psychiatric and cognitive affection of a group of medicated patients with FM and can be used as an adjunctive or can substitute medical treatment. 

## Figures and Tables

**Figure 1 brainsci-14-00416-f001:**
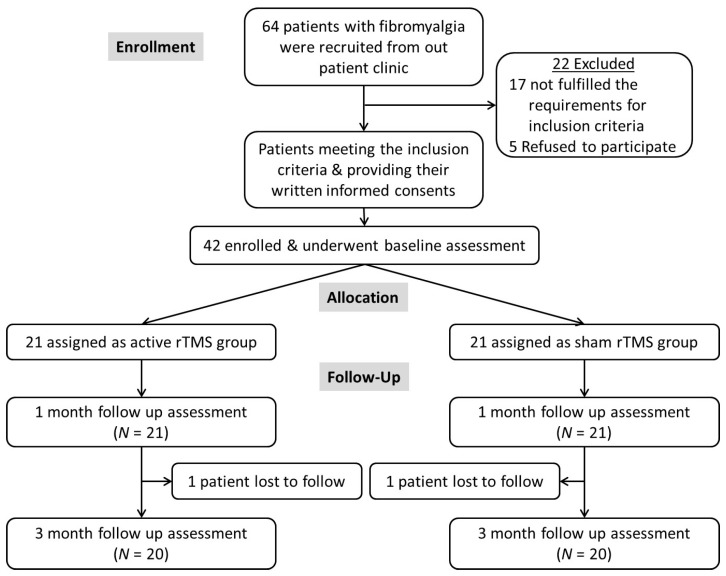
Shows the total number of recruited patients with fibromyalgia and the excluded patients, allocation, and follow-up after rTMS.

**Figure 2 brainsci-14-00416-f002:**
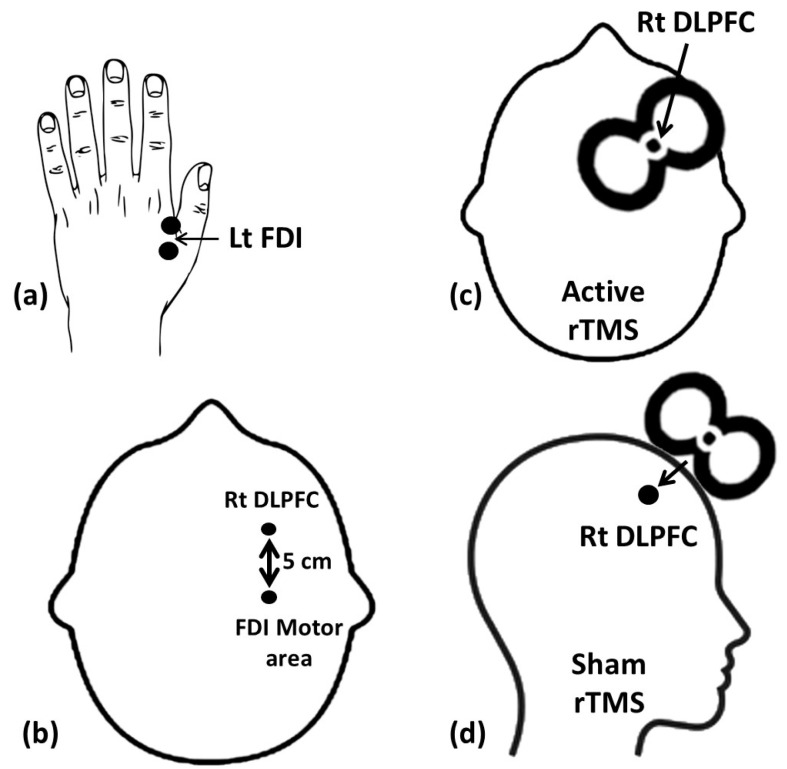
Location of stimulation point and position of TMS coil in real and sham groups. (**a**) First dorsal interossei muscle recording site; (**b**) location of the right dorsolateral prefrontal cortex; (**c**) position of the TMS coil in the active rTMS; (**d**) position of the TMS coil in the sham rTMS.

**Figure 3 brainsci-14-00416-f003:**
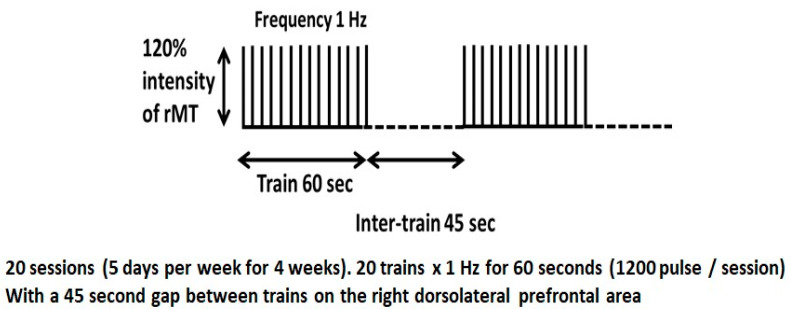
rTMS protocol: Each patient received 20 sessions (5 days per week for 4 weeks), with 20 trains × 1 Hz for 60 s with a 45 s gap between trains (total 1200 pulses/session) over the right dorsolateral prefrontal cortex.

**Figure 4 brainsci-14-00416-f004:**
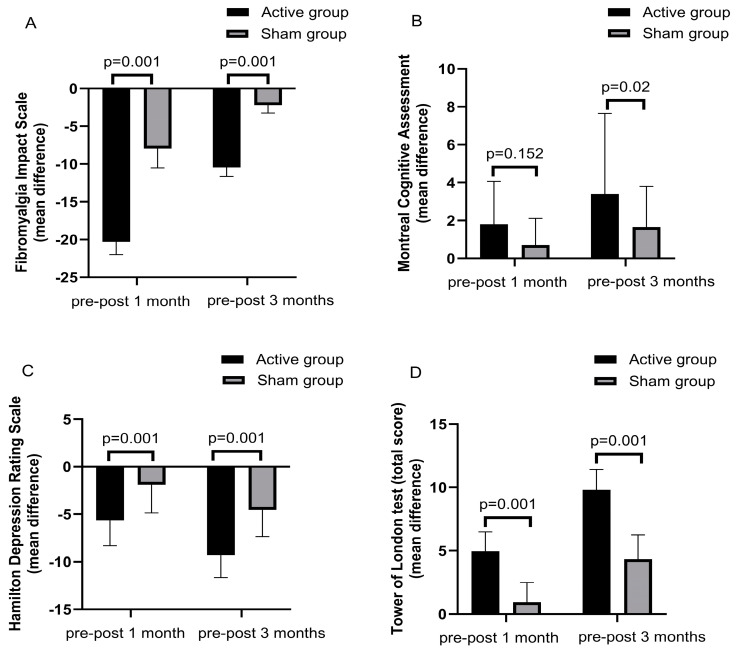
Mean changes (pre-post sessions and pre-post one month later and pre-post three months) in different rating scales. Significantly greater mean changes (more improvement) in the active group than the sham group were observed at both assessment points in the total scores of FIQ, MoCA, HDRS, and HARS.

**Table 1 brainsci-14-00416-t001:** Baseline assessment of studied groups regarding demographic and diagnostic aspects data.

	Sham Group (*n* = 20) (Mean ± SD)	Active Group (*n* = 20) (Mean ± SD)	Z	*p*-Value
Age	34.55 ± 8.32 34(14)	31.9 ± 7.49 31(10)	−1.002	0.316
Gender				
Males	8(40%)	9(45%)	0.102	0.5
Females	12(60%)	11(55%)		
Education years				
Less than 5 years	7(35%)	8(40%)	0.107	0.5
5 or more years	13(65%)	12(60%)		
Duration of illness Median IQR	6.60 ± 3.33 5.50(4.8)	8.30 ± 3.17 8.25(4)	−1.79	0.073
Fibromyalgia Impact Questionnaire (FIQ) Median IQR	58.05 ± 15.02 59.5(25)	63.25 ± 9.87 65(17)	−0.92	0.357
Hamilton Depression Rating Scale (HDRS) Median IQR	18.75 ± 2.40 19(5)	18.90 ± 2.19 19(4)	−0.205	0.838
Hamilton Anxiety Rating Scale (HARS) Median IQR	20.70 ± 1.92 20.5(3)	21.45 ± 2.01 21(3)	−1.094	0.274
Montreal Cognitive Assessment (MoCA) Median IQR	23.25 ± 1.83 23(4)	23.10 ± 2.04 23(4)	−0.205	0.837
Rey Auditory Verbal Learning Test (RAVLT) (sum of trials1–5 immediate real) Median (IQR)	27.4 ± 4.005 26.5(7)	26.9 ± 4.11 26(8)	−0.421	0.674
Rey Auditory Verbal Learning Test (RAVLT) (delayed recall) Median (IQR)	3.5 ± 1.05 3.5(1)	3.25 ± 1.33 3(2)	−0.980	0.327
Tower of London test (TOL) (total score) Median (IQR)	20.12 ± 1.84 20(3.1)	19.62 ± 1.74 19.85(2.8)	−0.758	0.448
Tower of London test (TOL) (solution time in seconds) Median (IQR)	27.28 ± 5.79 26.5(10.5)	27.35 ± 5.81 26.5(10.5)	−0.41	0.968
Tower of London test (TOL) (execution time in seconds) Median (IQR)	14.01 ± 4.17 14.5(7.8)	14.17 ± 4.04 14.5(7.8)	−0.204	0.839
Trail Making Test (TMT) part A Median (IQR)	70.5 ± 7.22 70(10)	69.7 ± 5.84 70(10)	−0.108	0.914
Trail Making Test (TMT) part B Median (IQR)	112.8 ± 8.49 113(14)	112.75 ± 8.84 113(16)	−0.027	0.978
Digit Span Test (DST) forward Median (IQR)	5.11 ± 0.67 5.15(1.1)	5.09 ± 0.59 5.05(1.1)	−0.149	0.882
Digit Span Test (DST) Backward Median (IQR)	4.52 ± 0.625 4.55(1.1)	4.40 ± 0.648 4.45(1.2)	−0.569	0.570

**
*Mann–Whitney test, IQR: interquartile range, SD: standard deviation.*
**

**Table 2 brainsci-14-00416-t002:** Medical treatment of the studied groups.

	Sham Group (*n* = 20)	Active Group (*n* = 20)	X^2^	*p*-Value
TCA and pregabalin/gabapentin	0(0%)	5(25%)	6.091	0.107
SSRI and pregabalin/gabapentin	6(30%)	5(25%)		
SNRI and pregabalin/gabapentin	10(50%)	6(30%)		
TCA, SSRI/SNRI, and pregabalin/gabapentin	4(20%)	4(20%)		

**
*TCA: tricyclic antidepressant, SNRI: serotonin and norepinephrine reuptake inhibitors, SSRI: selective serotonin reuptake inhibitors, Chi-square test.*
**

**Table 3 brainsci-14-00416-t003:** Differences between the mean changes in the Fibromyalgia Impact scale (FIQ) and the Hamilton Depression (HDRS) and Anxiety Rating Scales (HARS) at baseline, one and three months after the end of sessions among the studied groups.

Variable Assessment	Group I (Active rTMS) *n* = 20	Group II (+Sham rTMS) *n* = 20	*p*-Value	Group I (Active rTMS) *n* = 20	Group II (+Sham rTMS) *n* = 20	*p*-Value
	Mean Difference ± SD (Pre–Post 1 Month after Sessions)	Mean Difference ± SD (Pre–Post 1 Month after Sessions)		Mean Difference ± SD (Pre–Post 3 Month after Session)	Mean Difference ± SD (Pre–Post 3 Months after Sessions)	
Fibromyalgia Impact Questionnaire Median (IQR)	−20.30 ± 1.71 20(2.5)	−7.95 ± 2.56 8(2)	Z = −5.44, *p* = 0.001	−10.45 ± 1.19 10.5(1)	−2.2 ± 1.05 2(1)	Z = −5.47, *p* = 0.001
Hamilton Depression Rating Scale (HDRS) Median (IQR)	−5.65 ± 2.66 −6(1.75)	−1.9 ± 2.97 0(4)	Z = −3.228, *p* = 0.001	−9.30 ± 2.36 −10(4)	−4.55 ± 2.81 −6(3)	Z = −4.616, *p* = 0.001
Hamilton Anxiety Rating Scale (HARS) Median (IQR)	−5.05 ± 2.87 −6(4)	−0.80 ± 2.3 0(1.5)	Z = −4.086, *p* = 0.001	−6.35 ± 2.81 −7(5.75)	−2.95 ± 2.32 −2.5(4.5)	Z = −3.498, *p* = 0.001

**
*Mann–Whitney test, IQR: interquartile range, SD: standard deviation.*
**

**Table 4 brainsci-14-00416-t004:** Difference between the mean changes (pre–post 1 and 3 months after the end of sessions) of different cognitive rating scales among studied groups.

Variable Assessment	Group I (Active rTMS) *n* = 20	Group II (Sham rTMS) *n* = 20	*p*-Value	Group I (Active rTMS) *n* = 20	Group II (Sham rTMS) *n* = 20	*p*-Value
	Mean Difference ± SD (Pre–Post 1 Month)	Mean Difference ± SD (Pre–Post 1 Month)		Mean Difference ± SD (Pre–Post 3 Month)	Mean Difference ± SD (Pre–Post 3 Months)	
Montreal Cognitive Assessment (MoCA) Median (IQR)	1.8 ± 2.26 0(4.5)	0.7 ± 1.41 0(1)	Z = −1.43, *p* = 0.152	3.4 ± 4.25 4(4.25)	1.65 ± 2.15 0(3.5)	z = −2.27, *p* = 0.023
Rey Auditory Verbal Learning Test (RAVLT) (sum of trials1–5 immediate real) Median (IQR)	7.3 ± 3.09 6(3.25)	1.95 ± 3.88 0(2)	Z = −4.46, *p* = 0.001	12.15 ± 2.85 11(2)	4.9 ± 4.74 6(2.75)	Z = −4.50, *p* = 0.001
Rey Auditory Verbal Learning Test (RAVLT) (delayed recall) Median (IQR)	2.7 ± 0.80 3(0)	0.4 ± 0.94 0(0)	Z = −5.08, *p* = 0.001	5.7 ± 0.80 6(0)	1.55 ± 1.46 2(3)	Z = −5.62, *p* = 0.001
Tower of London test (TOL) (total score) Median (IQR)	4.95 ± 1.53 5(1.75)	0.95 ± 1.54 0(2)	Z = −5.04, *p* = 0.001	9.8 ± 1.6 10(2.5)	4.35 ± 1.89 5(2.42)	Z =−5.40, *p* = 0.001
Tower of London test (TOL) (solution time in seconds) Median (IQR)	−16.75 ± 6.11 −14.5(11.13)	−15.68 ± 5.46 −14.5(4.45)	Z = −0.286, *p* = 0.775	−10.8 ± 5.81 −12.5(7.5)	−3.5 ± 7.27 0(5)	Z = −2.60, *p* = 0.009
Tower of London test (TOL) (execution time in seconds) Median (IQR)	−3.57 ± 4.52 0(8)	−2.41 ± 4.34 0(4.38)	Z = −1.008, *p* = 0.314	−5.22 ± 4.36 −6.5(8)	−3.41 ± 4.61 0(8)	Z = −1.306, *p* = 0.192
Trail Making Test (TMT) part A Median (IQR)	−8.05 ± 9.02 −2.5(20)	−1.45 ± 2.78 0(2)	Z = −2.09, *p* = 0.037	−17.2 ± 6.03 −19(4)	−5.05 ± 7.33 −2(13.25)	Z = −4.11, *p* = 0.001
Trail Making Test (TMT) part B Median (IQR)	−6.6 ± 8.77 0(16)	−0.6 ± 1.42 −0(0)	Z = −2.146, *p* = 0.032	−12.35 ± 10.2 −15.5(20.5)	−4.3 ± 6.64 −1.5(4)	Z = −2.098, *p* = 0.036
Digit Span Test (DST) forward. Median (IQR)	2.13 ± 0.41 2(0)	1.1 ± 0.52 1(0)	Z = −4.98, =0.001	4.26 ± 0.54 4.2(0.75)	2.08 ± 0.46 2(0)	Z = −5.50, *p* = 0.001
Digit Span Test (DST) Backward Median (IQR)	2.57 ± 0.58 2.75(1)	0.45 ± 0.48 0.25(1)	Z = 5.52, *p* = 0.001	4.57 ± 0.58 4.75(1)	1.35 ± 0.46 1(1)	Z = −5.54, *p* = 0.001

**
*Mann–Whitney test, IQR: interquartile range, SD: standard deviation.*
**

**Table 5 brainsci-14-00416-t005:** Correlation between the changes in FIQ (changes in pre–post session, pre–post 3 months, and changes in psychiatric and cognitive scales among the real group.

	Changes in FIQ (Pre and Post Sessions)		Changes in FIQ (Pre and Post 3 Months)	
Hamilton Depression Rating Scale (HDRS)	r	−0.270	r	0.170
	*p*-value	0.250	*p*-value	0.474
Hamilton Anxiety Rating Scale (HARS)	r	0.235	r	0.214
	*p*-value	0.319	*p*-value	0.366
Montreal Cognitive Assessment (MoCA)	r	−0.436	r	0.089
	*p*-value	0.055	*p*-value	0.708
Rey Auditory Verbal Learning Test (RAVLT) (sum of trials 1–5 immediate recall)	r	0.536	r	0.315
	*p*-value	0.015	*p*-value	0.177
Rey Auditory Verbal Learning Test (RAVLT) (delayed recall)	r	−0.257	r	−0.062
	*p*-value	0.274	*p*-value	0.796
Tower of London test (TOL) (total score)	r	0.445	r	0.425
	*p*-value	0.049	*p*-value	0.062
Tower of London test (TOL) (solution time in seconds)	r	−0.283	r	−0.258
	*p*-value	0.227	*p*-value	0.273
Tower of London test (TOL) (execution time in seconds)	r	−0.149	r	−0.340
	*p*-value	0.530	*p*-value	0.142
Trail Making Test (TMT) part A	r	−0.011	r	−0.021
	*p*-value	0.963	*p*-value	0.930
Trail Making Test (TMT) part B	r	0.059	r	−0.220
	*p*-value	0.805	*p*-value	0.352
Digit Span Test (DST) forward	r	0.236	r	0.441
	*p*-value	0.317	*p*-value	0.052
Digit Span Test (DST) backward	r	−0.213	r	−0.067
	*p*-value	0.367	*p*-value	0.779

**
*Spearman’s correlation.*
**

## Data Availability

All data generated or analyzed during this study are available from corresponded on request. The data are not publicly available due to privacy/ethical restrictions.

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
