# Peer review of "Impact of Repetitive Transcranial Magnetic Stimulation on Cognitive and Psychiatric Dysfunction in Patients with Fibromyalgia: A Double-Blinded, Randomized Clinical Trial"

_brainsci, 2024, doi:10.3390/brainsci14050416_

Round 1
Reviewer 1 Report
Comments and Suggestions for Authors
The authors have focused on an important topic.
Page 2, under methods, line 77, :"The diagnosis of fibromyalgia was made according to American College of Rheumatology criteria or other standard criteria[26, 27]. Inclusion criteria: Participants aged ≥ 18 years".
Then, in page 4, line 146, "forty-two fibromyalgia patients who had been treated with appropriate medical treatment for at least 3 months received a total of 20 sessions (30 minutes each) of .."
If these patient's medications were continued, please mention that in inclusion.
Page 12, line 329, under conclusion, "patients diagnosed with FM had a broad spectrum of FIQ, psychiatric and cognitive abnormalities"
If these patients were on medications like Duloxetine or other pain medication, please provide that information in a table. It is important to know if these medications were continued or tapered and discontinued before the TMS study.
Author Response
For Reviewer 1
Thank you very much for your kind response and highly appreciated comments to improve the quality of the research. The next few lines introduce the response to your comments.
Comments to the Author
Page 2, under methods, line 77, :"The diagnosis of fibromyalgia was made according to American College of Rheumatology criteria or other standard criteria[26, 27]. Inclusion criteria: Participants aged ≥ 18 years". Then, in page 4, line 146, "forty-two fibromyalgia patients who had been treated with appropriate medical treatment for at least 3 months received a total of 20 sessions (30 minutes each) of "If these patient's medications were continued, please mention that in inclusion.
Thank you for your comments. The statement is corrected as follow:
"The diagnosis of fibromyalgia was made according to American College of Rheumatology criteria [26]. Inclusion criteria: Participants aged ≥ 18 years, who have been on their medication for the last 3 months without any improvement and express dissatisfaction. These medical treatments were continuing without changes along the time of the study. Also, we added table 2 shows medication treatment of the studied groups.
Page 12, line 329, under conclusion, "patients diagnosed with FM had a broad spectrum of FIQ, psychiatric and cognitive abnormalities”.
Thank you for your comments. The statement was corrected as follow: Patients diagnosed with FM had a broad spectrum of symptoms including; fatigue, psychiatric and cognitive abnormalities.
If these patients were on medications like Duloxetine or other pain medication, please provide that information in a table. It is important to know if these medications were continued or tapered and discontinued before the TMS study.
Thank you for your comments. We added table 2 shows medication treatment of the studied groups. These medical treatments were continuing without changes along the time of the study.

Reviewer 2 Report
Comments and Suggestions for Authors
This work aimed to evaluate the effect of low frequency rTMS over the right dorsolateral prefrontal area (DLPFC) on Fibromyalgia Impact Questionnaire (FIQ), psychiatric and cognitive disorders. 42 eligible patients with fibromyalgia (FM) were randomized to have 20 sessions of active or sham rTMS (1 Hz, 120% of resting motor threshold with total 1200 pules /session) over right 17 DLPFC.
The manuscript is well structured and deals with a topic of great relevance and potential interest for the scientific community. However, I have some suggestions for authors.
In particular, the introductory text is too concise. Transcranial magnetic stimulation. starting from 1985, and then especially in the last 10 years it has been widely used to carry out brain connectivity and neuromodulation studies.
To implement the introductory paragraph the authors can take into consideration the following recent publications:
- Moscatelli et al., Acute non invasive brain stimulation improves performances in volleyball players. Physiol Behav. 2023 Nov 1;271:114356. doi: 10.1016/j.physbeh.2023.114356;
The methods section is well structured. However, I believe it is appropriate to include a paragraph describing the experimental design in detail.
It is necessary to specify all acronyms used in the tables and figures in the relevant captions.
The paragraph of the discussions is too brief. Furthermore, it should begin by highlighting the main results of the study and, subsequently, discuss them based on what is present in the literature. Furthermore, the strengths of the study and its limitations should be specified in detail in this section. Finally, it would be appropriate to provide clear conclusions in which to also include possible future implications.
Author Response
For Reviewer 2
Thank you very much for your kind and honor reply. The authors are highly grateful to you. The authors highly appreciate your view about the paper. Your appreciated comments are considered as follows.
Comments to the Author
In particular, the introductory text is too concise. Transcranial magnetic stimulation. starting from 1985, and then especially in the last 10 years it has been widely used to carry out brain connectivity and neuromodulation studies. To implement the introductory paragraph the authors can take into consideration the following recent publications: - Moscatelli et al., Acute non invasive brain stimulation improves performances in volleyball players. Physiol Behav. 2023 Nov 1;271:114356. doi: 10.1016/j.physbeh.2023.114356;
Thank you for your valuable suggestion. We added the following in the introduction section:
In the last 10 years TMS has been widely used to carry out brain connectivity and neuromodulation studies.
Additionally, rTMS of DLPFC plays a special executive attention role in actively preserving access to stimulus representations and objectives in environments with plenty of distraction such as those of team sports. Volleyball is a team sport in which the attention and coordination components are essential for achieving performance (Moscatelli, Monda et al. 2023, Moscatelli, Toto et al. 2023). They concluded that rTMS of DLPFC improve reaction time (RT), the number of correct answer and the cortical excitability in Volleyball players after active rTMS and can increase coordination performances when the velocity of the execution is high. The added references were listed in the references list
The methods section is well structured. However, I believe it is appropriate to include a paragraph describing the experimental design in detail.
We added the paragraph in the methods section and added figure 2 and 3 to describe location and position of coil and describe the experimental design.
Experimental design: Each patient received 20 trains of low frequency (1Hz) rTMS at 120% of rMT of the left FDI, each train consists of 60 seconds with a 45 second intertrain interval giving a total of 1,200 pulses for each session (5 sessions / week, for 4 consecutive weeks) (see figure 3).
It is necessary to specify all acronyms used in the tables and figures in the relevant captions.
Thank you for your valuable comments. We revised and added it.
The paragraph of the discussions is too brief. Furthermore, it should begin by highlighting the main results of the study and, subsequently, discuss them based on what is present in the literature. Furthermore, the strengths of the study and its limitations should be specified in detail in this section. Finally, it would be appropriate to provide clear conclusions in which to also include possible future implications.
Thanks for your comment. All the discussion paragraphs are re written with highlighting the main results of the study and, subsequently, discussion based on what is present in the literature.
The strengths of the study was added, limitation and conclusion all were added to the study as follow:
The strengths of the study:
The number of treatment sessions (20 sessions) is the greater than any previous study were done on FM patients. The assessment scales of this study including impact of fibromyalgia on quality of life (FIQ), and both psychiatric and different cognitive assessment for somewhat long time (3 months) were carried out by 2 qualified neuro-psychiatric consultants blindly to the type of treatment sessions and the meaning of each score was taken.
Limitations of the study
The small number of patients studied and the short duration of the follow up were major limitations of this study. We recommend further studies with a larger sample size and for long-term follow-up.
- Conclusions
Patients diagnosed with FM had a broad spectrum of symptoms including fatigue, psychiatric and cognitive abnormalities. Application of 20 sessions of 30 minutes of low frequency rTMS over right DLPFC proved to be effective and safe for psychiatric and cognitive affection of group of medicated patients with FM and can be used as an adjunctive or can substitute medical treatment.
Many Thanks

Reviewer 3 Report
Comments and Suggestions for Authors
Dear authors,
Thanks for the submission. This article demonstrated that the 4 weeks of continuous rTMS could dramatically improve the multiple cognitive rating scales even after 1 and 3-month post-stimulation including but not limited to TOL, RAVLT, etc. This study might provide a new insight of using rTMS as an adjunctive or can substitute medical treatment in fibromyalgia patients.
However, some issues still need to be further addressed and revised.
1: There is a typo in the corresponding author’s email address. (line 11)
2: Flowchart figure 1, the fonts to be the same.
3: Should have a demonstration figure showing the precise region of where the stimulation site for both sham and stimulation groups. (especially for sham group)
4: In Table 1, the baseline assessment, should add in tower test of London data, trail making test and digit span test data in the table.
5: There is a huge improvement in both groups of Fibromyalgia Impact Scale (FIQ), Hamilton Depression (HDRS), and Anxiety Rating Scales (HARS), etc. even in the sham group, what is the possibility of such a positive impact in the sham group.
Author Response
Reviewer 3
I wish to thank you all for your constructive comments; your comments provided valuable insights to refine its contents and analysis.
Comments to the Author
1: There is a typo in the corresponding author’s email address. (line 11)
Thank you for your valuable comments. We revised it.
2: Flowchart figure 1, the fonts to be the same.
Flowchart figure 1 is corrected
3: Should have a demonstration figure showing the precise region of where the stimulation site for both sham and stimulation groups. (especially for sham group)
Thank you for your valuable comments. We added figure 2.
4: In Table 1, the baseline assessment, should add in tower test of London data, trail making test and digit span test data in the table.
Thank you for your valuable comments. We added it in table 1.
5: There is a huge improvement in both groups of Fibromyalgia Impact Scale (FIQ), Hamilton Depression (HDRS), and Anxiety Rating Scales (HARS), etc. even in the sham group, what is the possibility of such a positive impact in the sham group.
However, the observation of a significant improvement in the outcomes measured by the FIQ, HDRS, and HARS even in the sham or placebo group is not unusual in clinical trials, particularly those involving subjective measures or conditions with a strong psychological component. It may be related to the following: The test-retest effect refers to the phenomenon where participants' scores on a measure may improve or change simply due to the repeated administration of the same test or measure over time, regardless of any intervention or treatment. This effect can occur due to several reasons: Practice effect as participants may become more familiar with the test leading to improved performance or different responses on subsequent administrations, increased self-awareness as the act of completing the same measure multiple, and the familiarity with the testing process as the participants may feel more comfortable and less anxious during subsequent testing sessions, which could impact their responses or symptom reporting. (Hausknecht, Halpert, Di Paolo, & Moriarty Gerrard, 2007; Heilbronner et al., 2010, Scharfen et al 2018).

Round 2
Reviewer 2 Report
Comments and Suggestions for Authors
The authors responded to all my comments comprehensively